# Effect of Weld Length on Strength, Fatigue Behaviour and Microstructure of Intersecting Stitch-Friction Stir Welded AA 6016-T4 Sheets

**DOI:** 10.3390/ma16020533

**Published:** 2023-01-05

**Authors:** Dominik Walz, Robin Göbel, Martin Werz, Stefan Weihe

**Affiliations:** Materials Testing Institute (MPA), University of Stuttgart, Pfaffenwaldring 32, D-70569 Stuttgart, Germany

**Keywords:** friction stir welding, friction stitch welding, weld length, strength characteristics, fatigue strength, EN AW-6016, AA6016

## Abstract

Friction stir welding is a promising joining process for boosting lightweight construction in the industrial and automotive sector by enabling the weldability of high-strength aluminum alloys. However, the high process forces usually result in large and heavy equipment for this joining method, which conflicts with flexible application. In order to circumvent this issue, a friction stir welding gun has been developed which is capable of producing short stitch welds—either stand-alone as an alternative to spot welds or merging into each other appearing like a conventional friction stir weld. In this study, the influence of the stitch seam length on the strength properties of intersecting friction stir welds is investigated, and the weld is characterized. For this purpose, EN AW-6016 T4 sheets were welded in butt joint configuration with varying stitch lengths between 2 and 15 mm. Both the static and dynamic strength properties were investigated, and hardness and temperature measurements were carried out. The results show a scalability of the tensile strength as well as the fatigue strength over the stitch seam length, while the substitute proof strength is not affected. Hereby, the tensile strength reached up 80% of the base materials tensile strength with the chosen parameter setup. Likewise, the stitch weld length influences the hardness characteristics of the welds in the transition area.

## 1. Introduction

Aluminum and aluminium alloys are being increasingly used as materials for sustainable lightweight applications in the mechanical engineering, aviation and automotive sector to reduce CO${}_{2}$ emissions [1,2]. In particular, the 6XXX aluminum series is widely used for automotive applications such as closure panels, battery trays and body-in-white-parts due to their good strength properties and dent resistance [3].

A major challenge involved with fusion welding is the joining of both similar and foreign materials without losing the outstanding strength properties due to hot cracks occurring with high-strength aluminum alloys with a wide solidification range. The currently valid Eurocode 9 standard specifies strength losses of up to 40% for Metal Inert Gas (MIG) welds of the 6XXX-series [4]. In addition, other commonly used fusion welding technologies as laser welding still struggle with macro and micro cracks resulting from the melting of the material [5] or process stability issues like with resistance spot welding (RSW) [6]. To circumvent these challenges in fusion welding of high-strength aluminum alloys, the friction stir welding process, which was invented in the UK in 1990, is being increasingly used. This process operates below the solidus line and joins materials in solid state, which makes it possible to weld high-strength aluminum alloys with outstanding strength properties and comparatively inexpensive equipment.

The process can basically be divided into three steps, which are shown in Figure 1. The force, deformation and heat required for welding are provided by a rotating tool, which is pressed into the workpieces, penetrates them with a welding pin and thereby plastizises the material. The welding pin on the tool provides the stirring of the plasticized material, while the so-called tool shoulder provides the main part of the frictional heat. The tool is usually tilted by a few degrees to the vertical. This allows the workpieces to slide under the tool through the resulting gap and be compacted behind the tool to prevent the formation of pores.

In friction stir welding, the mechanical properties of the weld can be strongly influenced by the main parameters axial force, rotational speed and feed rate, as well as by the tool’s tilting angle. In addition, the parameters of the tool, especially the diameter of the shoulder and the pin geometry, play a major role in achieving good weld properties. The ratio of shoulder to pin diameter greatly affects the the tensile strength of the weld [7,8,9], and different or excentric pin profiles can further improve the mechanical properties and microstructure [10,11,12,13].

In order to apply the advantages of friction stir welding in car body production, a wide variety of production equipment and process variations exist. Mazda [14] and Kawasaki Heavy Industries [15], for example, use friction stir spot welding (FSSW) guns with C-frame for the production of tailgates and rear doors. In order to avoid the tool’s imprint in the spot weld or at the end of a weld in visible areas, these can be shifted to the back of the sheets to be joined either by switching the anvil and spindle positions [16] or by using segmented tools to fill them [17]. The weld strength of friction stir spot welded joints manufactured with monolithic tools is comparatively low due to the small welded area and segmented tools tending to wear [18] and require frequent maintenance [19]. Therefore, various attempts were made to increase the weld area for FSSW by an additional movement of the tool. This resulted in the so-called *friction swing welding* and *friction stitch welding.*

To be able to categorize these modifications to the FSW and FSSW process regarding the mechanical properties, various investigations were carried out for weld characterization and strength tests. Okamoto and Hunt investigated the friction stitch- and swing welding carried out on a self developed welding gun with thin sheets out of EN AW 6022-T4 [20,21]. Their stitch-welded lap-shear specimen showed increased shear strength with an increasing stitch length of up to 2.5 mm. In addition, the static and fatigue strength of the swing-welded specimens was significantly improved compared to RSW and FSSW spot welds. Tweedy et al. verified this correlation between weld size and static strength with swept FSSW specimens on EN AW-2024 and EN AW-7075 alloys in overlap configuration and for both like-kind and mixed joints [22]. The improvement of static strength properties of swept FSSW welds over conventional FSSW welds was verified also for EN AW-6061-T6 [23].

The novelty of this study is based on the focus on the intersecting area of short friction stir welds, carried out using a linear kinematic and the insights into the microstructure of EN AW-6016 welds through repeated thermal cycling. At present, there is no information in the literature describing or investigating stitch-welded FSW joints, especially not in butt configuration. All examinations which are described in the previous paragraph were carried out using a swing or gyrate motion to create an enlargement of the weld area. The described stitch welds were approximated from a swing motion with a distant pivot point [24]. Furthermore, no information could be found on the effect of intersecting stitch welds on the mechanical properties.

In this study, the weld characteristics and mechanical properties of intersecting stitch FSW welds in butt configuration were investigated. With a self-developed friction stir welding gun using a linear kinematic reported in [25], such welds can be produced on a flexible robot system. Figure 2 shows a comparison of the kinematics used in this study in contrast to the literature.

## 2. Materials and Methods

The materials used for this study were 2 mm thick EN AW-6016 T4 sheets. The elemental compositions and mechanical properties of the base material are shown in Table 1. The as-received sheets were cut into dimensions of 100 mm × 500 mm. The edges of the sheets were deburred and cleaned with ethanol before butt welding to prevent welding defects occurring from contamination (e.g., oil, dirt).

Welding was performed on an FSW machine by ESAB which can perform friction stir welds with an axial force up to 25 kN and a maximum feed speed of 4000 mm/min on one linear axis. To produce a friction stitch weld, a respective traverse length of the machine was set. After welding the first weld, the starting point was adjusted by adding the stitch length so that the plunging point of the following weld was located in the center of the exit hole of the previous weld. These intersecting friction stitch welds were repeated over the entire length of the workpieces (500 mm). For each welding pass, the stitch weld length was varied in steps of 2, 5, 10 and 15 mm. This resulted in a linear weld consisting of many intersecting stitch welds, as shown in Figure 3. The welding parameters shown in Table 2 were determined experimentally, and it was possible to achieve joints without any visible surface defects. A conventional friction stir weld, here referred to as *standard FSW*, was performed with the same parameter set for comparison. The FSW seam was oriented in the rolling direction of the aluminum sheets. During the welding process, the temperature was measured at six points near the starting point as seen in Figure 4 using K type thermocouples. The thermocouples were placed in a distance of 8, 10 and 12 mm, respectively, to the joint line.

After welding, the sheets were cut into stripes using guillotine shears and milled into the final specimen shape for tensile and fatigue testing according to [28] for tensile tests and ASTM E466 [29] for fatigue tests. Tensile tests were performed on a Zwick Roell 100 kN universal testing machine according to DIN EN ISO 6892-1, Method A1 at room temperature (20 °C). The strain was measured using a laser extensometer and a GOM Aramis adjustable 2D/3D system with a measuring length of 50 mm centered to the weld seam, respectively. The test speed was based on the strain speed of the laser extensometer. The stated values for the substitute proof strength $R_{p0.2}^{*}$ of the welded specimen are therefore valid only for this measuring length. Fatigute testing was performed on a Syncotec Powerswingly 20 kN using a periodic, sinusoidal forcing function. The stress amplitudes for the fatigue tests were determined using the bead string method according to Haibach [30] with a stress ratio of *R* = 0.1 (pulsating range) to prevent buckling of the specimens. The stress–life (S–N) curves presented in this study were fitted with the least square method and are each based on a minimum of 12 specimens; the tensile test results are based on 5 specimens, respectively.

The hardness measurements were carried out according to DIN EN ISO 6507-1 [31]. The microstructure analysis and microhardness measurements were performed in the intersecting area of the weld seams in order to be able to detect any defects resulting from re-entering of the tool. The specimens were given at least 4 months of natural ageing at room temperature after welding to obtain the same condition for the hardness measurements. The specimens were cut and polished using a cooling fluid to prevent a thermal influence on the microstructure. The polishing sequence was achieved with SiC abrasive paper in the following grit sizes: P180, P320, P500, P1000 and a subsequent fine polish with a diamond suspension in gradations of 15 µm, 6 µm and 1 µm. In a final step, the specimens were vibration polished with SiO${}_{2}$ and a grain size of 0.05 µm for approximately two hours before being etched with a hydrofluoboric acid (Barker etching) and examined under the light microscope (Leitz–Aristomet reflected light microscope, bright field). The Vickers hardness measurements were performed on a fully automated KB10 hardness testing machine.

## 3. Results and Discussion

### 3.1. Mechanical Properties

Figure 5 shows the substitute proof strength at 0.2% strain $R_{p0.2}^{*}$, ultimate tensile strength $R_{m}$ and the ultimate plastic elongation $A_{50}$ in each case averaged over all welded specimens and the base material for comparison. While the proof stress of all welded specimens is in the same area as the base material, the ultimate tensile strength increases with increasing stitch weld length, with the ultimate tensile strength for the 2 mm, 5 mm, 10 mm and 15 mm stitch welds reaching 184.5±3.2 MPa, 215.7±5.9 MPa, 232.5±5.8 MPa and 239.3±5.6 MPa, respectively. While the shortest stitch weld shows the lowest strength, it still reaches more than 67% of the base materials’ ultimate tensile strength. The ultimate plastic elongation scales with the stitch length as well but is significantly decreased in comparison with the base material, with the 2 mm stitch weld reaching 18.2% and the 15 mm stitch weld 50.3% of the base materials value.

The tensile properties of the standard FSW show highly decreased strength properties with high standard deviations compared to the other welded specimens and the base material, indicating an imperfect weld. These flaws are showed in Figure 6 in the form of periodic tunnel defects at the bottom of the weld.

#### 3.1.1. Fatigue Testing

Figure 7 shows the experimental results and normalized S-N lines of the fatigue tests. The point plots represent the fatigue life of the respective weld specimens at different stress amplitudes. Outliers are defined as those specimens that failed in the clamping zone. These are marked in the diagrams and crossed out.

For determining the finite life fatigue strength range, only specimens with a fatigue life between $N=10^{4}$ and $N=10^{7}$ cycles were used according to [32,33].

A least-squares fitting was used on the experimental data to determine the S-N line using the Basquin formula:(1)$$logN=k*log\;\sigma_{A}+c$$
with *k* being the negative slope of the S-N line:(2)$$k=\Delta \left(log\;N\right)/\Delta \left(log\;\sigma_{A}\right)$$

The inverse slope *k* varies slightly between the welded specimens, showing an increasing value with increasing stitch weld length and thus a flatter slope of the S-N line. This trend is also indicated by the evaluation of fatigue strength. The inverse slope values between 6 and 7 are in a good accordance with literature for friction stir welded specimens in butt configuration [34]. The R-squared value indicates a good fitting of the regression line for all stitch welds and the base material, deviating from this are the specimens of the continuous welds. Here, the influence of the weld imperfections is affecting the fatigue strength similarly to the static tensile tests. Due to the tunnel defects that occurred with this parameter set in the conventional FSW welds, the standard deviation increases significantly in both the tensile and the fatigue tests. Specimens with defects failed earlier; this is especially noticeable in fatigue tests, as here the tunnel defects can act as an internal notch from which an initial crack and subsequent failure originates. The fatigue strength was calculated from the S-N line at $N=2\times 10^{6}$ cycles and is shown in Table 3.

#### 3.1.2. Microhardness and Thermal Analysis

The Vickers hardness maps in cross-sections of the intersection point of two welds is shown in Figure 8 for each weld stitch length and the standard FSW, respectively. The cross section was taken from a re-entering point of two intersecting stitch welds and at a distance of 50 mm from the weld ends for the standard FSW. The hardness maps shown correspond in good accordance to the area which was affected by the shoulder of the tool. For all stitch welds, the crosssection can be divided into areas with different hardness values, which have altering shapes depending on the stitch length. The different hardness regions also match with the areas of the crosssection of a friction stir weld (nugget, thermomechanically affected zone (TMAZ), heat affected zone (HAZ) and base material). In the middle of the weld is an area of highly reduced hardness of 50 HV0.1 (TMAZ), which surrounds a smaller area with higher hardness values around 70 HV0.1 (nugget). At the sides of the greatly softened area, there are belts with moderate hardness values around 70–80 HV0.1 adjoin (HAZ), and, at the edges, the hardness increases up to the values of the EN AW-6016 T4 base material with 85–90 HV0.1. The 2 mm and 15 mm stitch welds are characterized by a comparatively small nugget and an extended TMAZ. The HAZ is similar in width for all stitch welds. The conventional FSW weld shows a nearly uniform hardness distribution with exceptions to the bottom of the weld. This is correlated to the tunnel defects in this specimen.

Since the increase of strength of the alloy EN AW 6016 is primarily based on precipitation hardening and, due to the comparatively high Si content, has both a high number of $Mg_{2}$Si precipitation phases and accelerated hardening kinematics [33], it can be expected that the short, cyclic heat input through repeated plunging into the weld leads to an influence of the number and size of the precipitates. According to Sato [35], the critical temperature at which accelerated grain and precipitate growth occurs during artificial ageing of FSW welds is above 175 °C. This temperature is repeatedly exceeded in the stitched weld specimens investigated in this work, with the maximum temperature in the first stitch weld depending on the length of the weld, as shown in Figure 9. Since the temperatures were measured at a distance of 8 mm, 10 mm and 12 mm from the joint, it can be assumed that the temperatures in the center of the weld are significantly higher, leading to faster hardening kinematics. If a heat input greater than 175 °C occurs repeatedly, as in the case of the 2 mm stitch weld, large precipitates grow at the expense of smaller ones [33]. The microstructure in the 2 mm stitch weld is starting to become over-aged, as indicated by a large softened zone. With decreasing cyclic heat input, this effect decreases.

Rodrigues investigated friction stir welds on 1 mm sheets out of EN AW 6016-T4. He also observed a decrease in hardness over the cross section of the weld with a parameter set he named “cold weld”. The parameter set with a higher heat input (“hot welds”) had the same hardness values as the base material [36].

The standard FSW has an almost uniform hardness distribution, which is in accordance with investigations by Sato and Rodrigues on friction stir welded AlMgSi alloys in the T4 heat treatment condition [35,36]. The softened zone at the bottom of the standard FSW aand the 15 mm stitch weld is due to a tunnel defect, which can be seen on the fractured tensile specimen in Figure 9.

These findings indicate that a parameter set which results in comparatively good strength properties for short intersecting stitched welds is not necessarily transferable to conventional friction stir welds. However, the extent to which the process parameters for intersecting stitch welds and the time intervals between the welds affect the material properties still needs further investigation.

## 4. Conclusions

The effect of the weld stitch length on the static strength, fatigue strength and hardness is investigated for intersected friction stitch welded 6016-T4 aluminum alloy in butt configuration.

The main results of this study are:It is possible to perform intersecting stitch welds with excellent strength properties and without flaws such as pores, kissing bond defects or tunnel defects.With the parameter set used in this study, visually defect-free intersecting welds up to a stitch length of 15 mm can be welded. For longer stitch welds or conventional friction stir welds, the welding parameters (dwelling time, plunging speed, rotational speed, feed speed) have to be adapted with regard to the time between the stitch welds.The tensile strength, ultimate plastic elongation and fatigue strength can be scaled with the stitch weld length. Specimens with a stitch weld length of 15 mm achieve 87% tensile strength and 50% ultimate plastic elongation of the base materials’ values. The shortest examined weld specimens reach 67% of the tensile strength of the base material due to the high cyclic heat input and the associated decrease in hardness. It should be highlighted that no influence of the weld stitch length on the substitute proof strength of the specimens could be investigated.The temperature cycles in intersecting stitched welds have a significant effect on the microstructure and hardness distribution of the welds. With decreasing stitch weld length, the cyclic heat input into the previous welds increases, resulting in an increased softened zone in the welds.

Future research will be conducted on lap joints with both naturally hard and age-hardenable aluminum alloys to investigate the applicability of the findings in this study. In addition, future specimens are planned to be welded with the self-developed friction stir welding gun in order to automate the stitch welds and eliminate the influences of human operation of the welding equipment. 

## Figures and Tables

**Figure 1 materials-16-00533-f001:**
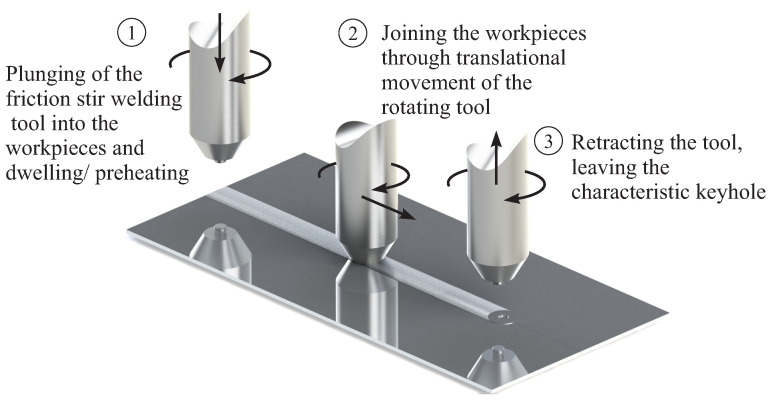
The three process steps of friction stir welding.

**Figure 2 materials-16-00533-f002:**
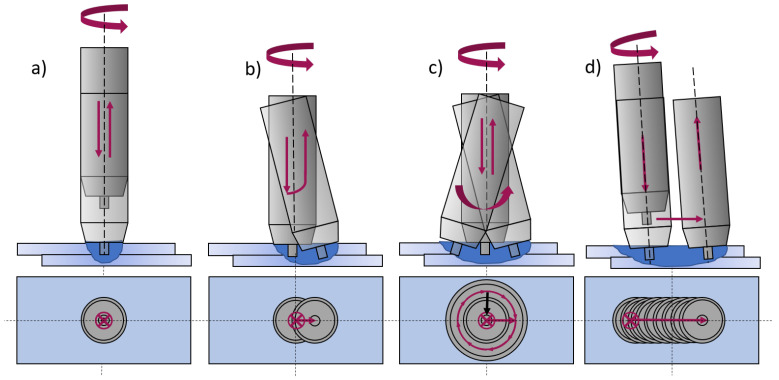
FSW and its modifications after [20,22]. (**a**) FSSW; (**b**) Swing-FSW; (**c**) Swing-FSW with a gyroidal movement; (**d**) Stitch-FSW.

**Figure 3 materials-16-00533-f003:**
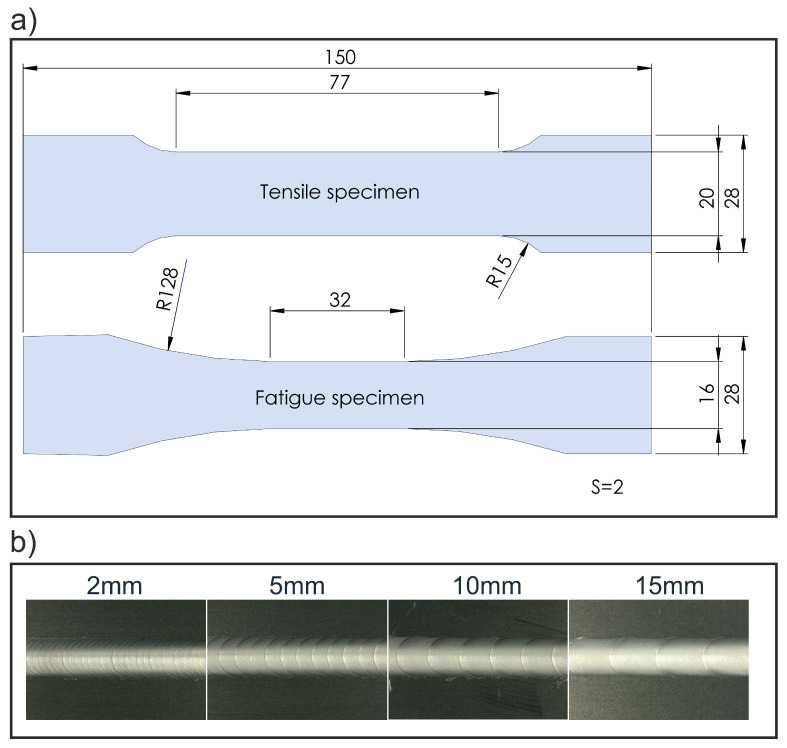
(**a**) Specimen geometry for tensile and fatigue testing according to DIN EN ISO 6892-1 and ASTM E466-15 [28,29] with all dimensions in millimeter.; (**b**) welded intersecting friction stitch welds with weld lengths between 2 mm and 15 mm.

**Figure 4 materials-16-00533-f004:**
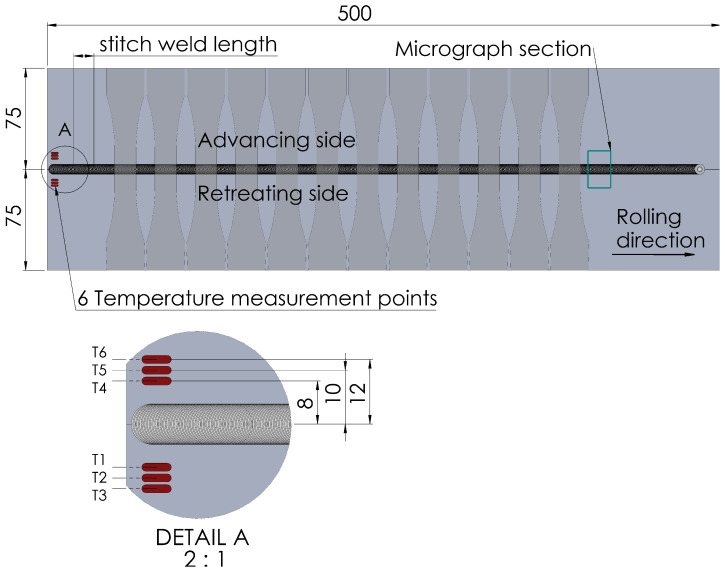
Specimen extraction from stitch welded sheets for tensile and fatigue testing, for micrographic analysis and measurement positions for thermocouples (all dimensions in millimeter).

**Figure 5 materials-16-00533-f005:**
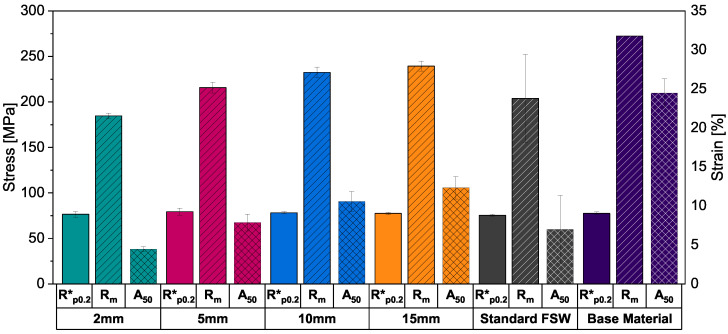
Tensile properties for all stitch weld lengths, the standard FSW and base material for comparison.

**Figure 6 materials-16-00533-f006:**
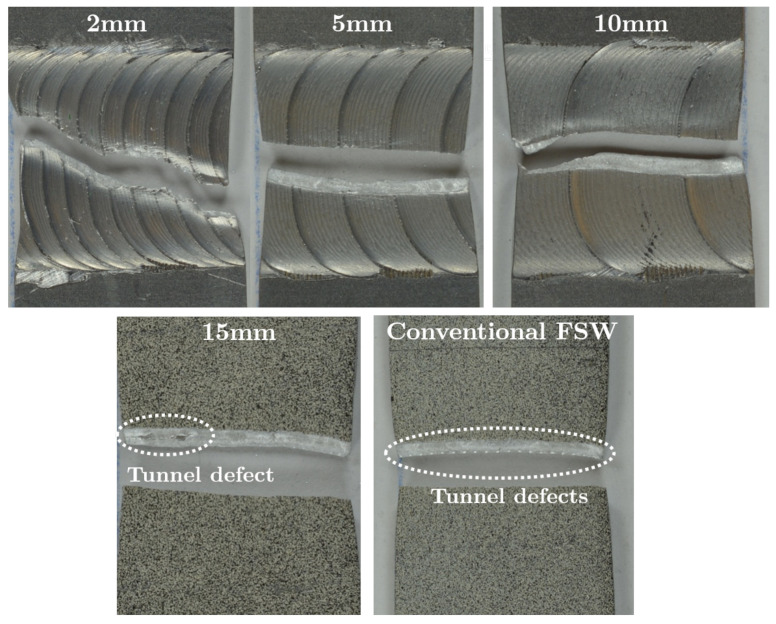
Fracture analysis of all stitch-welded specimen and the conventional FSW weld.

**Figure 7 materials-16-00533-f007:**
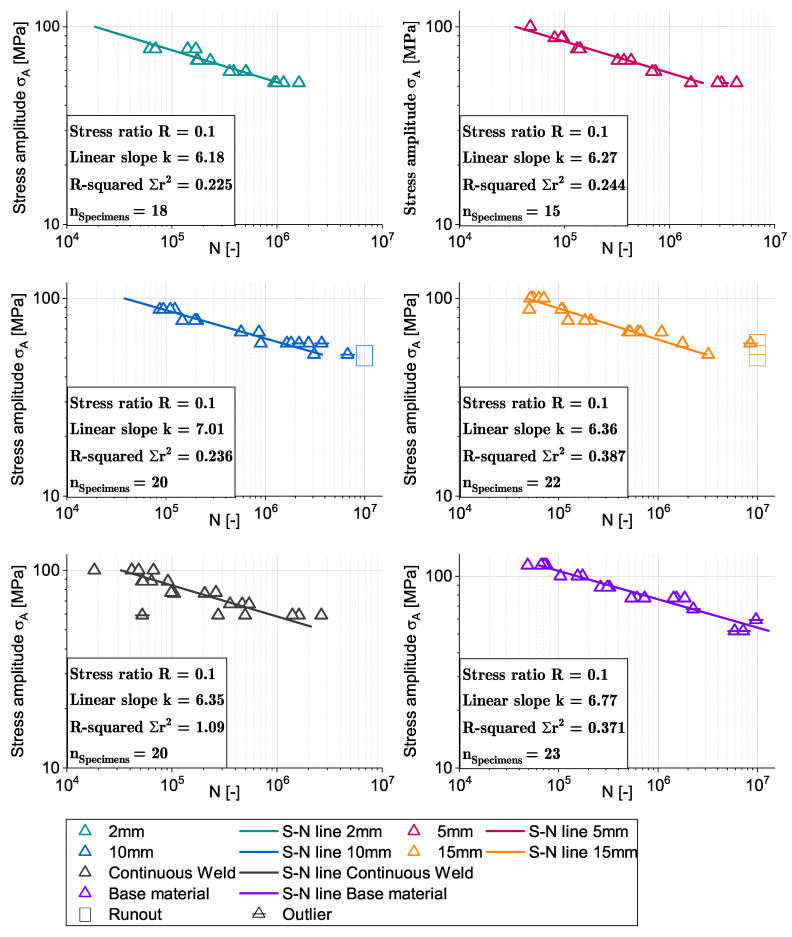
Experimental data of the fatigue tests and S-N lines for each stitch weld length, continuous weld and base material.

**Figure 8 materials-16-00533-f008:**
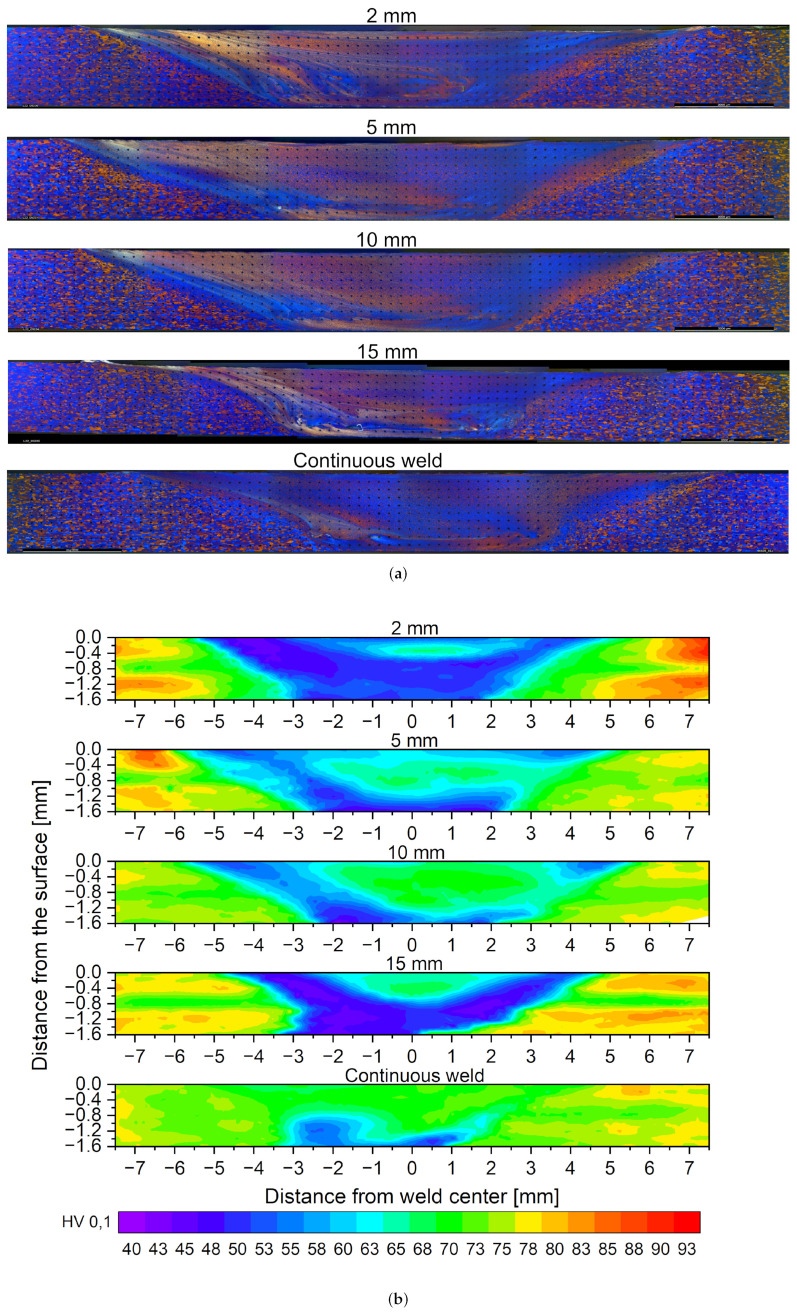
Results of the microscopic investigation and the hardness measurements. (**a**) crosssections of the stitch welds etched with Barker’s etchant and photographed under polarized light; (**b**) HV0,1 Hardness map of the cross sections of the stitch welds and the continuous weld.

**Figure 9 materials-16-00533-f009:**
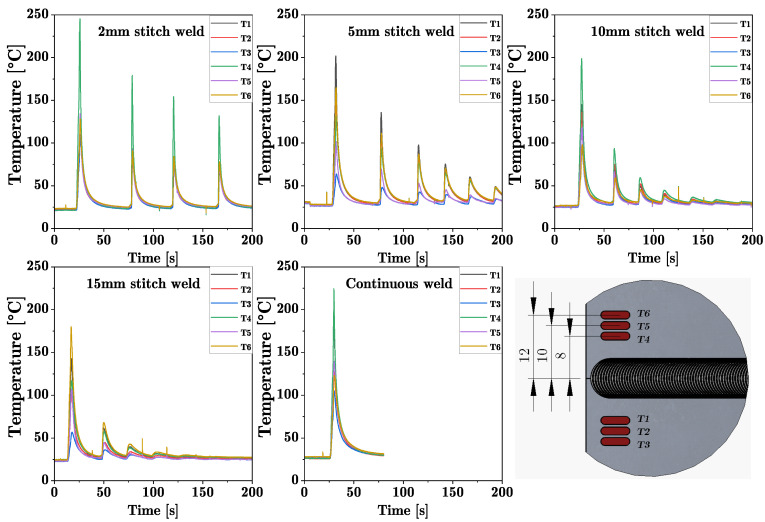
Thermocouple signals of all stitch welds in comparison to the continuous FSW and the positions of the thermocouples to the weld (all dimensions in millimeter).

**Table 1 materials-16-00533-t001:** Alloy composition of En AW-6016 T4 (AlSi1.2Mg0.4), according to DIN EN 573-3 and DIN EN 485-2 [26,27].

EN AW-6016 (AlSi1.2Mg0.4)	Si	Fe	Cu	Mn	Mg	Cr	Zn	Ti	Al
	[wt.%]	[wt.%]	[wt.%]	[wt.%]	[wt.%]	[wt.%]	[wt.%]	[wt.%]	[wt.%]
Chemical composition	1.0–1.5	0.5	0.2	0.2	0.25–0.6	0.1	0.2	0.15	Balance
	**Ultimate Tensile Strength** $R_{m}$ **[MPa]**		**Proof Strength** $R_{p0.2}$ **[MPa]**		**Total Strain at Maximum Load** $A_{\mathrm{gt}}$ **[%]**		**Hardness** **[HBW]**		
Mechanical properties	170–250		80–140		24		55		

**Table 2 materials-16-00533-t002:** Welding parameters and tool dimensions used for the stitch welds in this study.

	Rotational Speed	Forging Force	Feed Speed	Heel Plunge Depth	Tool Tilt Angle
	[1min]	[N]	[mmmin]	[mm]	[°]
Process parameters	1250	4850	1300	0.15	1
	**Shoulder Diameter**	**Pin Diameter**	**Pin Length**	**Shoulder Angle**	**Pin Structure**
	**[mm]**	**[mm]**	**[mm]**	**[°]**	**[-]**
Tool dimensions	14	5	1.8	10	No

**Table 3 materials-16-00533-t003:** Fatigue strength at $N\;=\;2\times 10^{6}$ cycles for all welded specimens and the base material.

Stitch weld length	2 mm	5 mm	10 mm	15 mm	Continuous weld	Base material
**Fatigue strength ** $\sigma_{A}$ **[MPa]**	46.88	52.28	56.84	55.55	52.38	68.54

## Data Availability

The data presented in this study are available on request from the corresponding author.

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
