# Peer review of "Effect of Weld Length on Strength, Fatigue Behaviour and Microstructure of Intersecting Stitch-Friction Stir Welded AA 6016-T4 Sheets"

_materials, 2023, doi:10.3390/ma16020533_

Round 1

Reviewer 1 Report

Manuscript ID Materials- 2086652 entitled " Effect of weld length variation on strength, fatigue behaviour and microstructure of intersecting stitch-friction stir welded AA 6016-T4 sheets" for journal of Materials has been reviewed.

- This article is comprehensive, logically organized and contains valuable information. However, there are few things need to be corrected and included in the manuscript for better understanding of carried research work to the readers.

+1- More references should be added to the introduction section. (especially about different studies)

+2- The novelty of the study should be further explained. (in introduction…)

+3- More information on tensile testing should be given in the " Materials and Methods " section. (humidity, ambient temperature, tester photo, etc.) The dimensions of the samples should be given in 3D (or tech. drawing).

+4- In tensile tests…. How was the crosshead speed determined? (please explain?)

+5- Microhardness assessments should be enriched more specifically. (page 8, line 168….)

+6- …. The R-squared value indicates a good fitting of the regression line for all stitch welds and the base material, deviating from this are the specimens of the continuous welds. Here, the influence of the weld imperfectios is affecting the fatigue strength similarly to the static tensile tests…. (please explain more, why?)

+7- The resolution of Figures 4 (and magnify), 8 and 9 should be increased. The thickness of the lines in the graphic (fig. 9) should be increased. Colours are not clear.

+8- More literature studies should be added to the introduction and other sections (DOIs given below).

DOI-1  https://doi.org/10.1007/s13369-021-06243-w  (different studies)

DOI-2  https://doi.org/10.35193/bseufbd.1075980   (info for tensile test samples… humidity….etc.)

DOI-3  https://doi.org/10.26701/ems.989945  (info about different studies)

--------------------------------------------------------

* It will be ready for publication after the specified corrections.

** I want to see article after the revision.

-------------------------------------------------------

Congratulations to the authors.

I wish the authors success in their future academic studies.

Kind regards.

Reviewer 2 Report

In this manuscript, the authors proposed a novel variant of FSW, namely stitch-friction stir welding. They investigated the effect of weld stitch length the static strength, fatigue strength and hardness. The findings of this manuscript are interesting, but it is lack of theoretical analysis. Therefore, a major revision must be performed, and the following issues must be addressed:

1.      The title can be revised into Effect of weld length on strength, fatigue behaviour and microstructure of intersecting stitch-friction stir welded AA 6016-T4 sheets.

2.      The authors give a good summary of the background of friction stir welding and some its variations. Apart from this, the authors are suggested to present a brief summary of the influencing factors in friction stir process. for example, the welding tool feature like shoulder to pin ratio, pin surface feature and even the pin eccentricity. So far, numerous researchers have studied the above factors. Some of the recommended papers to refer to are listed below:

https://doi.org/10.1007/s00170-022-09793-x

https://doi.org/10.1016/j.jmrt.2022.09.097

https://doi.org/10.1016/j.matpr.2019.12.042

3.      More detailed working principle of stitch-friction stir welding should be provided in the Introduction or Materials and Methods section.

4.      From my point of view, the process of the 15mm-stitch welding is more similar to a continuous FSW than 2mm, 5mm and 10 mm -stitch welds, but it is confusing that the thermal cycle curve of the 15mm-stitch weld is far different from that of continuous FSW weld, especially the peak temperature.

5.      The detailed parameters for hardness measurement were not given, including the applied load and dwell time.

6.      Figure 9 show that the continuous FSW has a higher processing temperature, but in Figure 8, the stitch welds were more softened?

Round 2

Reviewer 1 Report

Manuscript ID Materials- 2086652 entitled " Effect of weld length variation on strength, fatigue behaviour and microstructure of intersecting stitch-friction stir welded AA 6016-T4 sheets" for journal of Materials has been reviewed.

The authors have revised the manuscript carefully and the revised version could be published in the journal.

Decision- Accept

-----------------------------------------------------------------------

Congratulations to the authors.

I wish the authors success in their future academic studies.

Kind regards.

Reviewer 2 Report

The revised manuscript can be accepted now.